# Potency Assessment of Dendritic Cell Anticancer Vaccine: Validation of the Co-Flow DC Assay

**DOI:** 10.3390/ijms22115824

**Published:** 2021-05-29

**Authors:** Silvia Carloni, Claudia Piccinini, Elena Pancisi, Valentina Soldati, Monica Stefanelli, Anna Maria Granato, Toni Ibrahim, Massimiliano Petrini

**Affiliations:** 1Immuno-Gene Therapy Factory, IRCCS Istituto Romagnolo per lo Studio dei Tumori (IRST) “Dino Amadori”, 47014 Meldola, Italy; silvia.carloni@irst.emr.it (S.C.); claudia.piccinini@irst.emr.it (C.P.); elena.pancisi@irst.emr.it (E.P.); valentina.soldati@irst.emr.it (V.S.); monica.stefanelli@irst.emr.it (M.S.); annamaria.granato@irst.emr.it (A.M.G.); 2Osteoncology and Rare Tumors Center, Immunotherapy, Cell Therapy and Biobank, IRCCS Istituto Romagnolo per lo Studio dei Tumori (IRST) “Dino Amadori”, 47014 Meldola, Italy; toni.ibrahim@irst.emr.it

**Keywords:** dendritic cells, potency, T cell proliferation, validation, co-stimulation, flow cytometry

## Abstract

For many years, oncological clinical trials have taken advantage of dendritic cells (DC) for the design of DC-based cellular therapies. This has required the design of suitable quality control assays to evaluate the potency of these products. The purpose of our work was to develop and validate a novel bioassay that uses flow cytometry as a read-out measurement. In this method, CD3+ cells are labeled with a fluorescent dye and the DC costimulatory activity is measured by the degree of T cell proliferation caused by the DC–T cell interaction. The validation of the method was achieved by the evaluation of essential analytical parameters defined by international guidelines. Our results demonstrated that the method could be considered specific, selective, and robust. The comparison between measured values and estimated true values confirmed a high level of accuracy and a lack of systematic error. Repeated experiments have shown the reproducibility of the assay and the proportionality between the potency and the DC amount has proven its linearity. Our results suggest that the method is compliant with the guidelines and could be adopted as a quality control assay or batch-release testing within GMP facilities.

## 1. Introduction

Since their first description in 1973 [1], dendritic cells (DCs) have been considered the most efficient specialized antigen presenting cells (APCs) [2,3], with a unique ability to initiate, coordinate, and regulate adaptive immune responses [4].

Actually, among all the APCs, only DCs have the capacity to induce a primary immune response towards inactive or resting naïve T lymphocytes. To do this, DCs can uptake, process, and present antigens on their cell surface, along with the necessary accessory and costimulatory molecules, while undergoing the maturation process.

Thus, the interaction between mature DCs (mDCs) and antigen-specific T cells is the trigger of antigen-specific immune responses [5,6]. When interacting with CD4+ helper T cells, mDCs may induce their proliferation, activation, and differentiation into different CD4+ T cell subsets [7,8] throughout a complex phenomenon influenced by DC-derived cytokines and their maturation state [9,10].

Given the central role of DCs in initiating immune responses and surveillance, investigators have theorized that DCs would serve as an ideal tool for boosting endogenous anti-tumor responses that can lead to the effective eradication of tumors [11,12,13]. The role of the immune system in eliminating tumors has been established in several studies [14,15,16,17,18]. Monocyte-derived DCs (Mo-DCs) have been so far the most commonly used in anticancer vaccine clinical trials. In this approach, CD14+ monocytes selected from peripheral blood mononuclear cells (PBMCs) are cultured ex vivo and induced to differentiate into immature DCs (iDCs). Subsequently, the iDCs are stimulated by exposure to the appropriate maturation stimuli and simultaneously loaded with tumor antigens. Finally, the antigen-loaded mDCs are then harvested and cryopreserved in aliquots until the thawing at each scheduled vaccination date. To evaluate the quality of Mo-DC treatment, the potency test of the final product has become imperative for batch-release of DC products.

As defined in the US Code of Federal Regulations, potency is the specific ability or capacity of a product to affect a given result [19]. Potency is a critical quality attribute of biological products that has been determined using several bioassays. In particular, for DCs the mixed lymphocyte reaction assay has served as a “gold standard” for evaluating their functional ability to induce T cell activation. Alternatively, in 2004 Shankar et al. developed a method named “COSTIM bioassay” based on scintillation counting, which is suitable as a quality control assay or lot-release testing. In this functional test, T cells are stimulated with a sub-optimal amount of anti-CD3 antibody, such that they remain unable to proliferate unless a source of co-stimulation is added to the culture [20]. The following year, the same authors also validated the COSTIM bioassay for DC potency [21].

The main disadvantages of measuring T cell proliferation by applying scintillation counting are the instability and dangerousness of radioactive pyrimidine base and the lack of the exact number of cells which have actually proliferated due to the semi-quantitative nature of the method.

The aim of the present work was to validate the Co-Flow DC assay: a COSTIM assay using flow cytometry data as the T cell proliferation read-out.

## 2. Results

### 2.1. Method Development

In accordance with Shankar et al., we first focused on the purity of T cell samples used for subsequent analysis. Flow cytometric analysis revealed that the percentage of magnetically isolated CD3+ T cells was always higher than 90%. Additionally, Mo-DCs were checked for purity and maturation phenotype. Purity was always reported to be ≥60% (ranging from 60 to 84%) and the phenotype of the mDCs was confirmed by flow cytometry using the following markers: HLA-DR (accepted cutoff value ≥ 60%, average = 91.6%), CD80 (accepted cutoff value ≥ 50%, average = 98.7%), CD83 (accepted cutoff value ≥ 40%, average = 90.6%), and CD86 (accepted cutoff value ≥ 60%, average = 98.8%).

In all our experiments, the positive control was represented by T cells treated with the mitogenic agent phytohemagglutinin-L (PHA-L) (Figure 1A). In order to assess the best treatment conditions, we tested scalar concentrations of PHA-L and two different incubation times. Our results demonstrated that the best T cell proliferation rate was obtained after incubation with PHA-L 5 μg/mL for 68 h (Figure 1B).

### 2.2. Method Validation

The validation of the method described in this paper was performed satisfying essential analytical parameters defined by the Guidelines of the International Conference of Harmonization [22] and in accordance with the US Food and Drug Administration guidance document [19]. All the experiments were performed conforming to Current Good Manufacturing Practices in our quality control department to prove the specificity, selectivity, accuracy, linearity, robustness, and precision of the method. 

#### 2.2.1. Specificity

Specificity is the ability to accurately measure the analyte of interest in the presence of other components. Since the DCs used in the tests were Mo-DCs, we wanted to show that the observed T cell proliferation was independent of the stimulation induced by other APCs eventually present in the co-culture. For this purpose, we evaluated the potency induced by varying the numbers of mDCs or monocytes per well. As shown in Figure 2A, results obtained demonstrated that at 5 × 10^4^ stimulator cells per well, monocytes were capable of inducing a slight T cell proliferation, but still four times lower than that induced by DCs. However, monocyte-induced potency became neglectable when we cultured them at 1 × 10^4^ cells per well, whereas DCs maintained their costimulatory functions at this concentration. For this reason, 1 × 10^4^ DCs per well was maintained throughout the following experiments. Our aim was also to demonstrate that the proliferation of CD3+ T lymphocytes during the Co-Flow DC assay was specifically due to the DC costimulatory ability. We evaluated this by adding to the COSTIM cultures different concentrations of an antibody cocktail directed against DC antigens with costimulatory functions, such as anti-CD54, anti-CD80, and anti-CD86. The results showed that the simultaneous antibody-mediated blockade of these antigens completely prevented T cell proliferation (Figure 2B), indicating that in absence of adequate interactions between DCs and CD3+ cells during the co-culture the proliferative capacity is inhibited.

#### 2.2.2. Selectivity

Selectivity is the quality of a response that can be achieved without interference from any other substance. Additionally, in this case we considered monocytes as an interfering agent, therefore we seeded in the same well DCs and monocytes in different proportions for a total amount of 1 × 10^4^ cells. Our data demonstrated that the potency was proportional to the percentage of DCs present in the well and consequently inversely correlated with the percentage of monocytes (Figure 3).

#### 2.2.3. Accuracy

Accuracy describes the degree to which the result of a measurement conforms to the correct value and provides information on the ability of the test to produce solid and real results. Unable to establish a true value, our purpose was to evaluate the agreement between the obtained values (intended as the number of proliferated events) and the estimated expected values. The accuracy of the test was evaluated after seeding different proportions of DCs and monocytes in the same well, as described above. The expected value is determined by the contribution exerted by the quantity of DCs and monocytes present in the co-culture, as described in Table 1. 

We calculated the accuracy for each co-culture condition and its average value that was lower than the established acceptance criterion of 10%. The same data were further elaborated to evaluate if our test was affected by a systematic error. As shown in Figure 4, in our bioassay we found a lack of constant or proportional overrated and underrated data compared to the expected values. For this reason we can certainly assert that the Co-Flow DC assay is affected by a random, but not systematic error.

#### 2.2.4. Linearity

Linearity of the method was demonstrated by testing scalar concentrations of DCs cultured with 1 × 10^5^ T cells and therefore different ratios of DCs and CD3+ T cells during co-culture. Linearity is expressed as r-squared (r^2^) calculated by linear regression and obtained by interpolation between the potency results and the corresponding concentration of DCs per well (Figure 5). The obtained r^2^ is >0.978 indicating that the potency is directly proportional to the DC amount in the culture.

#### 2.2.5. Robustness

The robustness of an analytical procedure represents the ability of the test not to undergo alterations determined by small, but deliberate variations of the method parameters. In this case, we evaluated the potency results after variations of OKT-3 concentration and incubation time. In particular, we tested three different concentrations of OKT-3 antibody (0.005 μg/mL ± 0.001) and three different co-culture times (68 h ± 1). In both cases the coefficient of variation (CV) was lower than the established acceptance criterion of 10% (7.1 and 7.22, respectively). 

#### 2.2.6. Precision

The precision of an analytical procedure expresses the degree of reproducibility of the test, and it can be distinguished as repeatability and intermediate precision (IP). The precision assessment involved the use of a single batch of DC and a single batch of CD3+ cells. In order to verify the repeatability of the test, we observed the potency variation between replicates within the same co-culture plate (intra-assay). Repeatability results showed CV values lower than 10% (Table 2). 

On the other hand, the evaluation of IP involved the execution of three test runs on the same day by the same operator (inter-assay), three test runs each on a different day by the same operator (inter-day), and the execution of the test by three different operators (inter-analyst). For all the above-mentioned tests, the IP remained within the established acceptance criterion (CV ≤ 20%) (Table 2). 

## 3. Discussion

Potency assays are key tools for evaluating the critical quality attributes of medicinal products. In this study, we described the development and validation of the Co-Flow DC assay: a practical in vitro potency test for Mo-DC anticancer vaccines based on the COSTIM bioassay already described by Shankar et al. Our assay is focused on the flow cytometric analysis of proliferating T cells in co-culture with mDCs and a sub-optimal amount of anti-CD3 antibody.

In respect to the COSTIM bioassay, the time scale of the assay described in this study was optimized to the use of a different detection method. Despite this, several changes have been introduced into the process, as the greater duration of the assay that allowed us to introduce an appropriate positive control increasing the reliability of the test. Another considerable implementation is the use of Annexin V assay for the evaluation of the cell viability before the cell seeding. This test supplies more exact viability values that are essential to guarantee the correct proportion between effectors and stimulators within the assay. Therefore, the application of flow cytometry analysis permits us to know the absolute number of proliferated cells and to perform simultaneous labeling of other cell markers with the aim to obtain more information about the samples. Finally, we applied an important improvement in terms of safety using a flow cytometry method which let us avoid the use of radioactive reagents that could be dangerous for the analysts. 

On the other hand, the main limitation of the Co-Flow DC assay is the staining of T lymphocytes. In particular, times and conditions of labeling with PKH67 adversely affect the quality of proliferation data and the viability of T cells. Moreover, the seeding of the correct number of cells must be performed very carefully; thus personnel should be trained and retrained.

The validation of the above-described cell proliferation assay for the measurement of Mo-DC potency according to international guidelines for pharmaceutical products showed that it was suitable for this purpose with acceptable levels of specificity, selectivity, accuracy, linearity, robustness, and precision. 

This potency assay paves the way for the study of the correlation between potency values and clinical results. In our opinion, the best potency assessment for DC cell therapies comes from multi-strategy approaches that simultaneously evaluate T cell proliferation, cytotoxicity, activation markers, and cytokine release. 

Moreover, specific inhibitory/costimulatory molecules would be added to the Co-Flow DC assay to evaluate their impact on T cell activity and proliferation.

In conclusion, the Co-Flow DC assay represents a new functional bioassay that will help us to better understand the relationship between the biological activity of cell therapy products and the clinical efficacy of the treatment improving our knowledge of DCs anticancer vaccines.

## 4. Materials and Methods

### 4.1. Isolation of PBMCs and Cell Purification

Human PBMCs were isolated from healthy donor’s blood by density gradient centrifugation using Lymphocyte Separation Media (Biowest, Nuaillé, France). Then, CD3+ T cells or CD14+ monocytes were obtained by magnetic separation, using the producer’s recommended protocol. In particular, CD3+ T cells were isolated by negative depletion of CD14+, CD15+, CD16+, CD19+, CD34+, CD36+, CD56+, CD123+, and CD235a+ cells, using the Pan T Cell Isolation Kit (Miltenyi Biotec, Bergisch Gladbach, Germany). While, CD14 MicroBeads (Miltenyi Biotec) were used for the positive selection of monocytes. The purity of isolated cells was checked by flow cytometry and only samples with purity > 90% were used for subsequent experiments. Isolated CD3+ and CD14+ cells were cryopreserved in 90% heat inactivated human serum AB (Biowest) and 10% dimethyl sulfoxide (DMSO; Mylan, Dublin, Ireland) solution until use.

### 4.2. DCs Culture

Mo-DCs were prepared from healthy donor-derived PBMCs, as already described [23]. PBMCs were cultured with CellGro DC Medium (CellGenix GmbH, Freiburg, Germany) at 10 × 10^6^ cells/mL for 2 h after which all cells that had not adhered to the plastic were removed from the culture. Adherent cells were cultured in CellGro DC Medium added with 1000 IU/mL of recombinant human (rh) interleukin (IL)-4 and rh-granulocyte-macrophage colony-stimulating factor (GM-CSF; CellGenix GmbH). On day 7 the culture medium was discarded and the cells were incubated in CellGro DC Medium added with IL-6 (2000 UI/mL), tumor necrosis factor-α (TNFα; 20 ng/mL), IL-1β (20 ng/mL) (all from CellGenix GmbH), and prostaglandin E2 (PGE2; 1 μg/mL) (Cayman Chemical, Ann Arbor, MI, USA). On day 9 mDCs were collected, washed, resuspended in sterile saline solution, and counted under a light microscope to assess their vitality and purity. DCs were cryopreserved in 90% autologous plasma and 10% DMSO solution until use.

### 4.3. Immunophenotypic Analysis

For the purity assessment of magnetically isolated cell populations, T cells were stained with anti-human Viogreen CD3 recombinant antibody (REA) (1:50; Miltenyi Biotec Cat# 130-113-704, RRID:AB_2726245), whereas monocytes were stained with anti-human Viogreen CD14 REA (1:50; Miltenyi Biotec Cat# 130-110-583, RRID:AB_2655056) for 10 min at 4 °C in the dark. Expression of Mo-DCs surface markers was measured by flow cytometry following induction to terminal differentiation using fluorescently conjugated REA against CD86 (1:50; Miltenyi Biotec Cat# 130-116-265, RRID:AB_2727438), CD80 (1:50; Miltenyi Biotec Cat# 130-123-314, RRID:AB_2802032), CD83 (1:50; Miltenyi Biotec Cat# 130-110-504, RRID:AB_2659323), and HLA-DR (1:50; Miltenyi Biotec Cat# 130-113-968, RRID:AB_2726435). Appropriate conjugated REA Controls (S) (1:50; Miltenyi Biotec) were included for each sample. Cells were washed twice in autoMACS running buffer before being analyzed by flow cytometry. The exclusion of dead cells from flow cytometric analysis was performed with 7-amino-actinomycin D (7-AAD) staining solution (Miltenyi Biotec Cat# 170-080-032) following the manufacturer’s instructions.

### 4.4. Co-Flow DC Assay

CD3+ cells and allogenic DCs were thawed, washed, and resuspended in warm AIM-V medium 1X (Gibco, Thermo Fisher Scientific, Waltham, MA, USA). A total of 2 × 10^5^ cells of each sample were incubated with 10 μL/mL Annexin V-FITC in binding buffer (eBioscience Annexin V-FITC Apoptosis Kit, Invitrogen, Carlsbad, CA, USA) for 15 min at 37 °C in a humidified atmosphere in the dark. Cells were then washed and suspended in binding buffer. Immediately before flow cytometric analysis, propidium iodide (PI) was added to a final concentration of 5 μg/mL to distinguish between total apoptotic cells (Annexin V+ and PI− or +) and necrotic cells (Annexin V- and PI+). T cells were labeled with PKH67 Green Fluorescent Cell Linker Midi Kit (Sigma Aldrich, St. Louis, MO, USA), as already described [24]. In a U-bottom 96-well plate, 1 × 10^4^ live DCs and 1 × 10^5^ live T cells were co-cultured in triplicate (background). Moreover, the same number of cells were seeded with 0.005 µg/mL OKT-3 (Prodotti Gianni, Milan, Italy) (COSTIM). In every plate a positive control consisting of T cells and phytohemagglutinin-L (PHA-L, Life Technologies, Carlsband, CA, USA) and a negative control consisting of T cells and OKT-3 (0.005 µg/mL) were included. The co-culture was performed for 68 h at 37° C in a humidified atmosphere. At the end of incubation, cells were harvested and analyzed by flow cytometry. As additional reagents for the assay validation, the monoclonal anti-human CD54 (Miltenyi Biotec Cat# 130-104-031, RRID:AB_2658701) and REA Control (S) (Miltenyi Biotec Cat# 130-104-616, RRID:AB_2661695) antibodies were added to COSTIM to prove the specificity of the method.

### 4.5. Flow Cytometry

Flow cytometry acquisition was carried out on the MACSQuant Analyzer 10 (Miltenyi Biotec) equipped with 405 (violet), 488 nm (blue), and 640 (red) lasers and 10,000 events were recorded for each sample. The acquisition and analysis gates were set on lymphocytes, monocytes, or DCs based on forward (FSC) and side scatter (SSC) properties of cells. FSC and SSC were set on a linear scale. Flow cytometry data were analyzed with MACSQuantify 2.13 Software (Miltenyi Biotec). Proliferation analysis was performed using Cell Tracking Wizard in Modfit LT 4.1 Software (Verity Software House, Topsham, ME, USA). All the analyses were executed setting the parental generation on the negative control and using the floating model. The percentage of proliferating cells was calculated subtracting the percentage of cells of the parental generation from 100.

### 4.6. Data and Statistical Analysis

For each Co-Flow DC experiment the average proliferation value of triplicate wells was estimated. The potency was calculated by subtracting the proliferation value of the background to the proliferation value of the COSTIM condition. The test was considered evaluable when the proliferation value of the background was between the positive control value and the negative control value. The variability of the potency results obtained throughout the method validation was evaluated by applying the percentage CV. For the comparison of data obtained from different analysts, we used the ICC based on absolute agreement, two-way mixed-effect model.

## Figures and Tables

**Figure 1 ijms-22-05824-f001:**
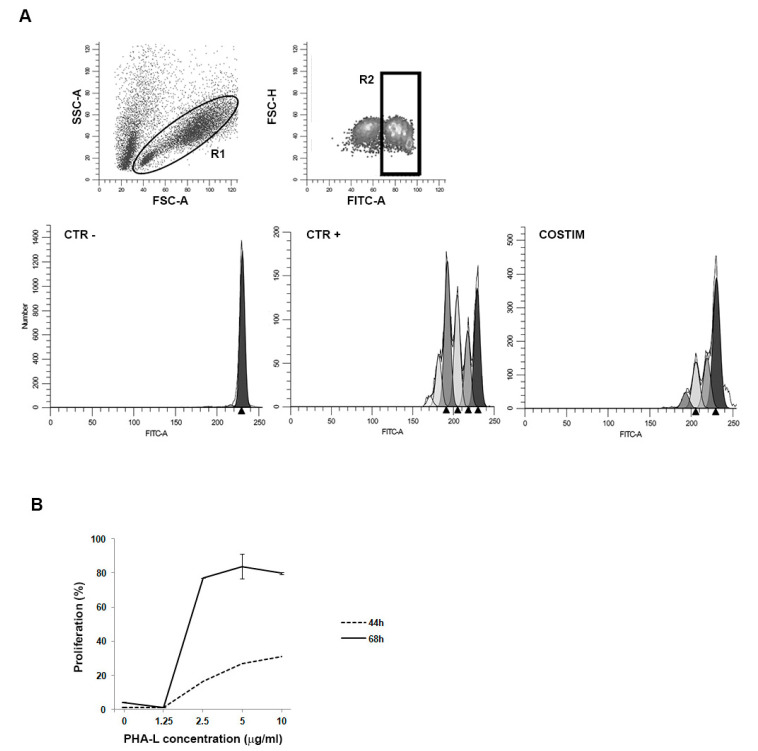
PKH67 stained CD3+ T cell proliferation. (**A**) Gating strategy and representative flow cytometric histograms analyzed with Modfit LT. COSTIM = co-culture of T cells + DCs + OKT-3. (**B**) Labeled T cells were co-cultured with different concentrations of PHA-L and analyzed by flow cytometry after 44 h (dashed line) or 68 h (continuous line) in culture. T cell proliferation results are mean ± standard deviation (SD) of three experiments at both incubation times.

**Figure 2 ijms-22-05824-f002:**
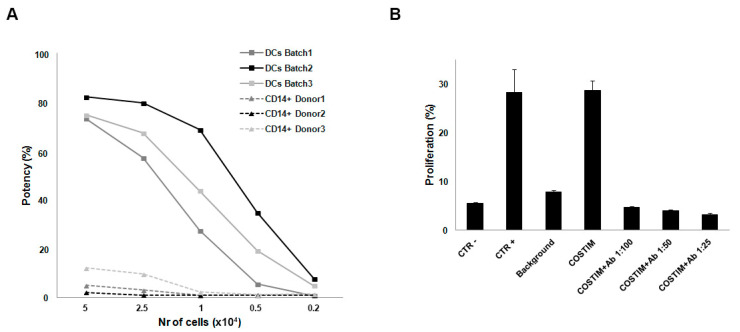
Specificity of the bioassay. (**A**) Potency induced by varying numbers of DCs or monocytes per well. Data from three batches and donors are reported. (**B**) Proliferation with and without different concentrations of the antibody (Ab) cocktail (anti-CD54, anti-CD80, and anti-CD86). Results are mean ± SD of three experiments. The addition of the isotopic controls does not modify the DC-induced T cell proliferation at any concentration.

**Figure 3 ijms-22-05824-f003:**
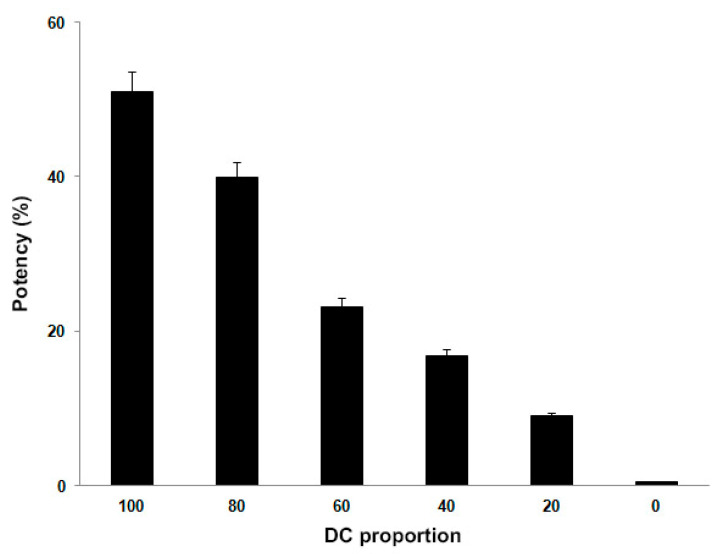
Selectivity of the bioassay. Potency data obtained adding various proportions of CD14+ cells to three different batches of DCs. Results are reported as mean ± SD.

**Figure 4 ijms-22-05824-f004:**
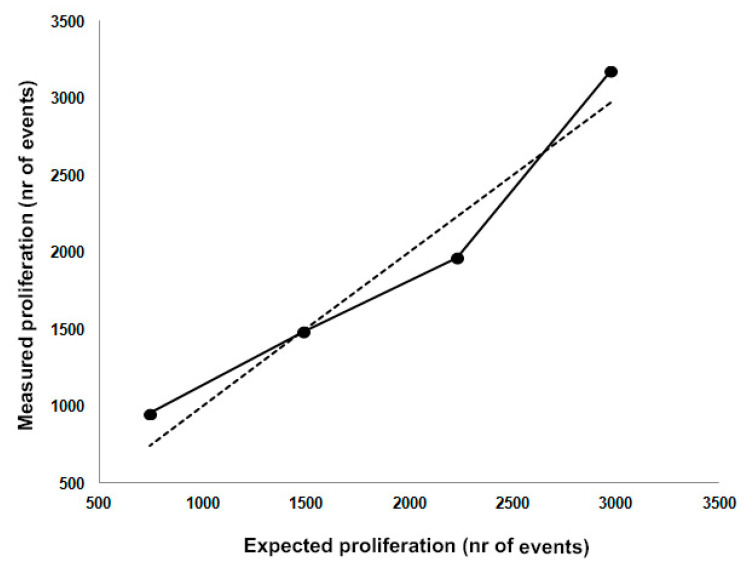
Definition of the experimental error. Continuous line represents the observed data in terms of the number of proliferated events. Dashed line represents the “true” values based on the calculated expected proliferation as reported in Table 1.

**Figure 5 ijms-22-05824-f005:**
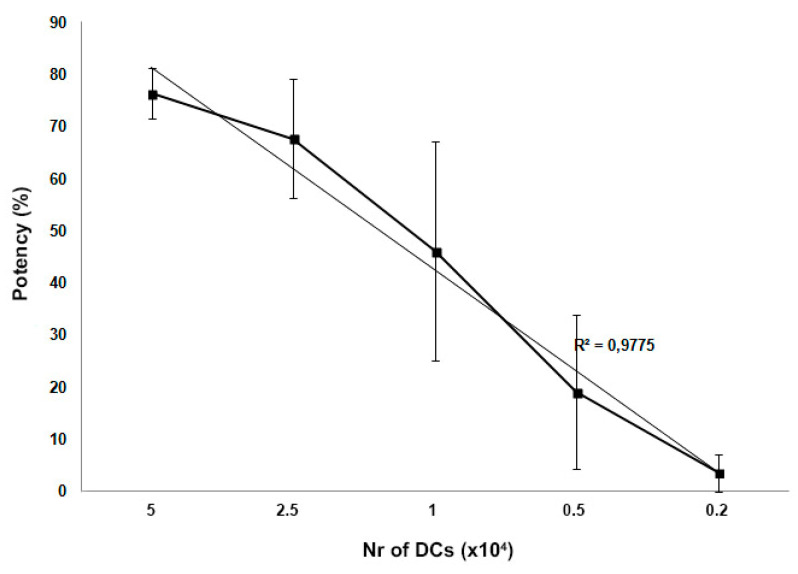
Linearity of the method. Dose response of varying numbers of DCs co-cultured with CD3+ T cells (1 × 10^5^) and its linear regression line are reported. Results are reported as mean ± SD obtained from three different batches of DCs.

**Table 1 ijms-22-05824-t001:** Accuracy and relationship between expected and measured values.

	Proportion (%) of DCs and Monocytes in Culture					
	100 + 0	80 + 20	60 + 40	40 + 60	20 + 80	0 + 100
Measured values (nr of proliferated events)	3715.2	3172.9	1965.1	1483.6	951.1	182.7
Expected values ^1^	-	2972.5	2229.8	1487.2	744.5	-
Accuracy (%)	-	6.7	−11.9	−0.2	27.8	-
Average accuracy (%)		5.6				

^1^ Expected value was calculated as the multiple of “proportion of DCs in culture” and “the measured value” obtained from the 100% DC group, plus the multiple of “proportion of monocytes in culture” and “the measured value” obtained from the 100% monocyte group.

**Table 2 ijms-22-05824-t002:** Evaluation of repeatability and intermediate precision.

	Run	CV (%)	ICC *	
	*n*	Mean (range)	Acceptance criterion	
Intra-assay	3	6.46 (0.93–9.78)	≤10	
Inter-assay	3	14.3	≤20	
Inter-day	3	6.16	≤20	
Inter-analyst 1 Inter-analyst 2	3	16.97 (9.04–23.54) 16.95 (5.01–31.29)	≤20	0.693 0.945

* Based on the 95% confident interval of the intraclass correlation coefficient (ICC) estimate, values less than 0.5, between 0.5 and 0.75, between 0.75 and 0.9, and greater than 0.90 are indicative of poor, moderate, good, and excellent reliability, respectively.

## Data Availability

The datasets generated and/or analyzed during the current study are available from the corresponding author on reasonable request.

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
