# Peer review of "Potency Assessment of Dendritic Cell Anticancer Vaccine: Validation of the Co-Flow DC Assay"

_ijms, 2021, doi:10.3390/ijms22115824_

Round 1

Reviewer 1 Report

The manuscript by Carloni et al. describes the validation of a novel co-flow potency assessment assay for dendritic cells (DC) for DC-based cellular therapies. The manuscript contains a set of novel interesting valuable data, is well written and easy to follow. I have only couple minor suggestions:

- I think it would be nice to summarize at one place the benefits and limitations of a novel method in comparison to existed ones. This summary could be supplied with a table or visualized graphically.

- In my opinion the title could be simplified.

Author Response

Response to Reviewer 1 Comments

Point 1: I think it would be nice to summarize at one place the benefits and limitations of a novel method in comparison to existed ones. This summary could be supplied with a table or visualized graphically.

Response 1: We are grateful to the referee for this useful comment. On the basis of the referee’s remark, we summarized the limitations of our assay in the Discussion. We didn't add the table in this section because we believe that it wouldn't have added any information to the text.

Point 2: In my opinion the title could be simplified.

Response 2: We agree with the referee’s comment, so we modified the title.

Reviewer 2 Report

Silvia Carloni et al. present in their manuscript a novel method how to reliably evaluate the potency of a dendritic cell vaccine against cancer. The concept of the vaccine is that extracted dendritic cells induce after extracorporeal activation with tumour antigens T-lymphocytes to eliminate tumour cells from the organism. 

They propose to measure the activity of the modified dendritic cells by measuring the proliferation of CD4+ - T-lymphocytes. The  T-lymphocyte population is measured after fluorescence labelling using flow-cytometry. This is a modified assay, introduced by Shankar et al. using radioactive T-cell labelling. 

The authors show in a convincing way that their measurement method is specific and selective. T-cells grow proportionally to the number of dendritic cells. The measurement method is robust in relation to varying the culture time and antibody concentration and it is reproducible with a small standard variation. 

The only evaluation method I don't agree with is the accuracy evaluation. A test for accuracy is a test that the measurement method measures what it is pretending to measure. For this an alternative approach to determine the factor to be measured is required. If this other method is not available (as stated by the authors) a relative accuracy can still be obtained. 

Such a relative accuracy measurement compares the relative obtained measurement value with the known relative concentration. That means - you obtain the measurement for one sample. Then you half the concentration of the measured agent by diluting it by a factor of 2 keeping all other parameters the same (here: keeping the total number of cells / millilitre constant). Your measurement value should be half of what it was before - and so on. Like this you determine whether the measurement behaves linearly with the concentration of T-cells, whether their is an offset (the curve does not go through the null-point) and what the measurement range is before the method saturates. 

The accuracy measurement proposed by the authors invests that there is a linear response of the number of T-cells and the number of dendritic cells after proliferation. This is an assumption which should not be part of the accuracy determination of a measurement method. 

If I misunderstood their method and the authors did precisely what I propose then they should reformulate this section and eliminate the word "proliferation". This word leads me (and possibly others) to the understanding that they actually let the cells proliferate before they take the measurements. 

Author Response

Response to Reviewer 2 Comments

Point 1: The only evaluation method I don't agree with is the accuracy evaluation. A test for accuracy is a test that the measurement method measures what it is pretending to measure. For this an alternative approach to determine the factor to be measured is required. If this other method is not available (as stated by the authors) a relative accuracy can still be obtained.  Such a relative accuracy measurement compares the relative obtained measurement value with the known relative concentration. That means - you obtain the measurement for one sample. Then you half the concentration of the measured agent by diluting it by a factor of 2 keeping all other parameters the same (here: keeping the total number of cells / millilitre constant). Your measurement value should be half of what it was before - and so on. Like this you determine whether the measurement behaves linearly with the concentration of T-cells, whether their is an offset (the curve does not go through the null-point) and what the measurement range is before the method saturates. The accuracy measurement proposed by the authors invests that there is a linear response of the number of T-cells and the number of dendritic cells after proliferation. This is an assumption which should not be part of the accuracy determination of a measurement method. 

If I misunderstood their method and the authors did precisely what I propose then they should reformulate this section and eliminate the word "proliferation". This word leads me (and possibly others) to the understanding that they actually let the cells proliferate before they take the measurements. 

Response 1: We are grateful to the referee for this comment and we regret not having been sufficiently clear in our explanation. In order to avoid future misunderstandings, we deleted the word "proliferation" from the section "Accuracy".